# Perinatal asphyxia and associated factors among neonates admitted to a specialized public hospital in South Central Ethiopia: A retrospective cross-sectional study

Seifu Awgchew Mamo[1☯], Girum Sebsibie Teshome[2☯], Tewodros Tesfaye[2☯], Abel Tibebu Goshu[3☯]*

1 Department of Pediatric Nursing, College of Medicine and Health Sciences, Wachemo University, Hosaena, Ethiopia, 2 School of Nursing and Midwifery, College of Health Sciences, Addis Ababa University, Addis Ababa, Ethiopia, 3 School of Nursing and Midwifery, College of Health and Medical Sciences, Haramaya University, Harar, Ethiopia

☯ These authors contributed equally to this work.
* ABTI2222@gmail.com

**Data Availability Statement:** All relevant data are within the paper.

## Abstract

### Introduction

Perinatal asphyxia continues to be a significant clinical concern around the world as the consequences can be devastating. World Health Organization data indicates perinatal asphyxia is encountered amongst 6–10 newborns per 1000 live full-term birth, and the figures are higher for low and middle-income countries. Nevertheless, studies on the prevalence of asphyxia and the extent of the problem in poorly resourced southern Ethiopian regions are limited. This study aimed to determine the magnitude of perinatal asphyxia and its associated factors.

### Methods

A retrospective cross-sectional study design was used from March to April 2020. Data was collected from charts of neonates who were admitted to NICU from January 2016 to December 31, 2019.

### Result

The review of 311 neonates' medical records revealed that 41.2% of the neonates experienced perinatal asphyxia. Preeclampsia during pregnancy (AOR = 6.2, 95%CI:3.1–12.3), antepartum hemorrhage (AOR = 4.5, 95%CI:2.3–8.6), gestational diabetes mellitus (AOR = 4.2, 95%CI:1.9–9.2), premature rupture of membrane (AOR = 2.5, 95%CI:1.33–4.7) fetal distress (AOR = 3,95%CI:1.3–7.0) and meconium-stained amniotic fluid (AOR = 7.7, 95% CI: 3.1–19.3) were the associated factors.

**Funding:** The authors received no specific funding for this work.

**Competing interests:** The authors have declared that no competing interests exist.

**Abbreviations:** SDG, Sustainable Development Goal; WHO, World Health Organization; WCSH, Worabe Comprehensive Specialized Hospital; SVD, Spontaneous Vaginal Delivery; ANC, Antenatal Care; NICU, Neonatal Intensive Care Unit; PROM, Premature Rupture of Membrane; DM, Diabetes Mellitus; CS, Cesarean Section; PICs, Pregnancy Induced Complications; PIHDs, Pregnancy Induced Hypertensive Disorders.

## Conclusion

Substantial percentages of neonates encounter perinatal asphyxia, causing significant morbidity and mortality. Focus on early identification and timely treatment of perinatal asphyxia in hospitals should, therefore, be given priority.

## Introduction

A decade remains to achieve the third Sustainable Development Goal (SDG) targets, including reducing preventable causes of neonatal mortalities [1, 2]. Though the number of neonatal deaths has declined significantly within the last thirty years, Africa alone accounted for about 41% of the total global neonatal mortality in the year 2017 [3].

Perinatal asphyxia, also called birth asphyxia, is one of the leading causes of neonatal deaths in the world following severe infections and prematurity [1, 4]. It results from the loss of the blood supply or impairment of gas exchange to or from the fetus before, during, or after the birth process [5]. Perinatal asphyxia may lead to severe metabolic acidosis, hypercarbia, progressive hypoxemia, neonatal encephalopathy, and multi-system organ failure, and even result in death [6–10].

An arterial blood sample, APGAR score (appearance, pulse, grimaces, activity, and respiration), immediate neurological complications, or evidence of multiple organ dysfunction are used to diagnose asphyxia. The APGAR score is assessed in the first and fifth minutes of life and ranges from zero to ten. According to the WHO classification, an APGAR score of four to seven in the first minute of life indicates moderate prenatal asphyxia, whereas zero to three suggests severe asphyxia [11, 12]. Factors related to antepartum, intrapartum or immediate postpartum period may contribute to perinatal asphyxia development [13]. These could be summarized into abnormal maternal oxygenation, congenital infections, insufficient placental perfusion, traumatic deliveries, or impaired umbilical circulation. Most studies identified that intrapartum factors as having the highest impact [7, 14, 15].

Worldwide, perinatal asphyxia is encountered amongst 6–10 newborns per 1000 live full-term birth [9, 16, 17]. Apparently, the numbers are higher for low and middle-income countries. In Africa, significant neonatal morbidity and mortality occur due to complications associated with perinatal asphyxia. Studies from different regions of the continent show variant figures on the magnitude. A study from two district hospitals in Ghana showed that birth asphyxia to be the second cause of admission (15.1%) and the third cause of mortality (20.7%). In contrast, another study conducted in Nigeria indicated severe perinatal asphyxia to be the most important cause of death in all birth weight categories except in extremely low birth weight babies [18, 19]. A systematic review conducted amongst east and central African countries has shown a pooled prevalence of perinatal asphyxia to be 15.9% [9].

Ethiopian studies are no different from the rest of Africa. Though the country has made efforts to minimize child and neonatal mortality rates by devising a National strategy for Newborn and Child Survival, current figures still show demand for committed action to target and tackle preventable causes like birth asphyxia [20, 21]. The Ethiopian Demographic and Health survey in 2019 revealed a reduction in early childhood mortality except for neonatal mortality, which had steady progress of decline since 2016. It was shown that birth asphyxia was and still is the primary cause of neonatal deaths [22]. Observational studies in different parts of the country also magnify this concern. A cross-sectional study conducted in the Tigray region of northern Ethiopia showed that 22.1% of the total neonates included in the study experienced

perinatal asphyxia [23]. At the same time, another study from Dilla referral hospital revealed the prevalence of perinatal asphyxia to be 32.8% [24].

In order to improve the quality of care that is delivered within a country, one must first understand the problem and the issues that healthcare providers encounter. Improving the care of the maternal-infant dyad can lead to a significant decrease in the rates of perinatal asphyxia [22, 25]. In an effort to enhance care in Ethiopia, we sought to understand the burden of perinatal asphyxia and early assessment practices. To do this, we evaluated cases of perinatal asphyxia and identified associated factors among neonates admitted to the neonatal intensive care unit of Worabe Comprehensive Specialized Hospital (WCSH) in southern Ethiopia.

## Methods

### Study area and period

The study was conducted at Worabe Comprehensive Specialized Hospital (WCSH), Silte zone, south central Ethiopia, from March to April 2020 (by reviewing neonates' charts from January/2016 to December 31/2019). WCSH was established in October 2014 and is located 170 km southwest of Addis Ababa. It is the only referral hospital in Silte zone and provides emergency, out-patient and in-patient services to the local and neighboring communities. The Neonatal Intensive Care Unit has 18 beds, 5 incubators, 15 radiant warmers and operates with 17 nurses, 2 midwives, 1 general practitioner, 2 pediatricians and 1 gynecologist. The NICU offers diagnostic and treatment services for approximately 1000–1500 newborns per year [26]. The level of newborn care is determined by the neonate's gestational age, birth weight, sickness severity, and the facility's general setup. The NICU service provision at WCSH is leveled into three. Accordingly, health practitioners at the level I offer basic newborn care to low-risk infants and triage unwell newborns. Specialty treatment (Level II NICU) is confined to newborn infants over 32 weeks gestational age weighing 1500 g or recovering from severe disease treated in a level III environment (subspecialty). All newborn infants with extreme preterm (28 weeks or less) or extremely low birth weight (1000g or less), or severe and/or complex disease are handled in level III NICU [26].

### Study design

A facility-based retrospective cross-sectional study design was employed.

### Population

**Source population.** All records' of neonates who were admitted at NICU of Worabe comprehensive specialized hospital.

**Study population.** All the records' of neonates who were admitted to the NICU of WCSH were added to the study. The records' of neonates with incomplete documentation (no proper maternal or fetal measurement parameters), or with major congenital malformations or anomalies were excluded.

**Sample size determination.** The sample size was calculated using the single population proportion formula considering the p-value of 32.8% from a previous study conducted in Ethiopia [24].

$$ni = (z\alpha/2)^2 * p(1 - p)$$

$$d^2$$

Where, $ni$ = initial sample size, p = proportion of prenatal asphyxia; 32.8% = 0.328, $(z\alpha/2)^2$ = confidence interval (95%), d = is the margin of sampling error tolerated (5%) = 0.05

Substituting the values for each of these variables in the above formula, the sample size was estimated to be 339.

Because the source population is less than ten thousand, we used a correction formula

$$nf = ni/(1 + ni/N)$$

Where nf = final sample size
ni = calculated sample size (initial sample size)
N = total number of neonates admitted to NICU of WCSH within the three years period (3796 neonates)

$$nf = 339/(1 + 339/3796) = 311$$

## Sampling procedure

Systematic random sampling technique was used to select which charts to review. After the records of the neonates were put in their order of admission, the $k^{th}$ interval was determined by dividing the total population size by the total sample size.

$$k = N/nf = 3796/311 = 12$$

Where, N is the total population at NICU of WCSH from January /2016 to December 31/ 2019,
nf = final sample size of the study.

## Operational definitions

Perinatal asphyxia–is the inability of a newborn to initiate and sustain respiration by persistently scoring an APGAR score of less than 7 for more than 5 minutes after delivery [24, 27].

Prolonged labor–is the first stage of labor, exceeding 12 hours in primigravida or 8 hours in multipara mothers [25].

Premature rupture of membranes (PROM)–rupture of the amniotic sac and chorion membrane occurred before the onset of labor [24].

## Study variables

Magnitude of perinatal asphyxia was the outcome variable. Maternal socio-demographic variable (age in year, educational status, religion, residency, and occupation), antepartum related variable (anemia during Pregnancy, ANC follow up, DM, pregnancy-induced hypertension, antepartum hemorrhage, chronic hypertension, gravidity, and parity), intrapartum related variable (PROM, prolonged labor, obstructed labor, place of delivery and mode of delivery), neonatal related variable (birth weight, fetal distress, the gender of the neonate and intrauterine meconium release) were independent variables.

## Data collection tools and procedures

Data was collected from neonates' registration and medical records using a structured checklist adapted and modified from different literature [13, 23, 24, 28]. The checklist contained four sections which assess socio-demographic, antepartum factors (parity/gravidity, gestational age, complications/illnesses during pregnancy), intrapartum factors (mode of delivery, duration of labor, fetal presentation, fetal distress, intrauterine meconium release occurrence of PROM, and obstructed labor) and neonatal factors (sex, birth weight, gestational age at

birth and occurrence of perinatal asphyxia). To identify maternal-related factors, the maternal record chart of the same year was traced. Three nurses with bachelor's degree were recruited as data collectors and were trained for two days on the checklist, data collection, and supervision. Furthermore, the sampling procedures were also elucidated to them. The checklist was pretested for its relevance and clarity to answer the research question.

### Ethical considerations

Ethical clearance for the commencement of the study was obtained from the Addis Ababa University, School of Nursing and Midwifery Research Ethical Committee (Ref. no. 030/20/ SNM). Permission was also sought and obtained from the ethical committee of Worabe comprehensive specialized hospital. Data was kept anonymous by keeping the identity of the neonate's or mother's credentials hidden before, during, and after the study.

### Data analysis and management

The data were entered into statistical software Epi-data version 3.1. The entered data was subject to cleaning using simple frequency and tabulation to ensure the data's validity. Then the analysis was made with IBM SPSS version 24.0 after exporting the prepared data. Descriptive statistics, such as frequency distribution, were computed to describe the significant variables of the study. Odds ratio and the p-value were computed. $P \leq 0.05$ was considered as statistically significant for association. Binary logistic regression was conducted to see the effect of each of the independent variables on the outcome variable, and variables that were statistically significant at P value less than 0.05 levels were put into the final model (multivariate analysis) to control for confounding.

## Results

### Socio-demographic characteristics of the mothers' of neonates

Data were collected from a total of 311 neonatal medical records. Of the neonates' mothers, 105(33.8%) were between the ages of 25–29 years, and 163(52.4%) of the mothers had no formal education. Almost all the mothers, 297 (95.5%), were Muslim by religion [**Table 1**].

### Obstetrics history of the mothers at WCSH

Regarding parity, 196(63%) mothers were multiparous, and 197(63.3%) of them were attending their antenatal follow up. Among the mothers, 129(41.5%) had anemia, while 86(27.7%) were diagnosed with preeclampsia. Only 61(19.6%) of mothers had gestational diabetes mellitus, and few mothers, 21(6.8%), had chronic hypertension. Of the 311 mothers, 92(29.6%) had an antepartum hemorrhage, and 222(71.4%) mothers of the neonates delivered by SVD. Concerning membrane status, 100(32.2%) mothers developed premature rupture of the membrane, and 115(37%) mothers had obstructed labor [**Table 2**].

### Clinical characteristics of neonates at WCSH

From the total number of neonates, 192(61.7%) of them were males, and 119 (38.3%) were females, and most of the neonates, 240(77.2%), had birth weight between 2.5kg to 4kg. Only 84 (27%) of babies passed their meconium during labor, and 119(38.3%) of them developed fetal distress [**Table 3**].

**Table 1. Socio-demographic characteristic of mothers of newborns at WCSH, South central Ethiopia (n = 311).**

| Variable | Category | Frequency/ Percentage of mothers' of neonates (n = 311) | Frequency/Percentage of mothers' of asphyxiated neonates (n = 128) |
|---|---|---|---|
| Age | ≤19yr | 12 (3.9%) | 6 (4.68%) |
| | 20-24yr | 65 (20.9) | 24 (18.75%) |
| | 25-29yr | 105 (33.8%) | 44 (34.37%) |
| | 30-34yr | 84 (27%) | 34 (26.56%) |
| | ≥35 | 45 (14.5%) | 20 (15.625%) |
| Educational status | No formal education | 163 (52.4%) | 78 (60.93%) |
| | Primary | 64 (20.6%) | 24 (18.75%) |
| | Secondary | 45 (14.5%) | 14 (10.93%) |
| | Higher education | 39 (12.5%) | 12 (9.37%) |
| Residence | Rural | 207 (66.6%) | 93 (72.65%) |
| | Urban | 104 (33.4%) | 35 (27.34%) |
| Religion | Muslim | 297 (95.5%) | 123 (96%) |
| | Orthodox | 10 (3.2%) | 3 (2.34%) |
| | Protestant | 4 (1.3%) | 2 (111.56%) |
| | Catholic | 0 (0%) | 0 (0%) |
| Occupation status | Housewife | 120 (38.6%) | 57 (44.53%) |
| | Private | 2 (0.6%) | 1 (0.78%) |
| | Government | 63 (20.3%) | 23 (17.96%) |
| | Merchant | 112 (36%) | 46 (35.93%) |
| | Student | 14 (4.5%) | 1 (0.78%) |

## Magnitude of perinatal asphyxia

The overall prevalence of perinatal asphyxia among neonates who were delivered at WCHS was found to be 128(41.2%).

## Factors associated with the occurrence of perinatal asphyxia

Different socio-demographic, neonatal, and maternal variables were tested for their association with the presence of perinatal asphyxia. For the multivariate logistic regression, all variables that were found to have an association with the outcome variable in bivariate logistic regression at P = 0.25 were included in the multivariate logistic regression models. During multivariate logistic regression, six variables (preeclampsia, antepartum hemorrhage, gestational diabetes mellitus, premature rupture of membrane, intrauterine meconium release, and fetal distress during labor) were found significantly associated with perinatal asphyxia (at p≤0.05).

Among maternal factors, preeclampsia was independently associated with significantly higher odds of the newborn developing perinatal asphyxia (AOR = 6.2, 95% CI, 3.1, 12.3). Neonates from mothers who had antepartum hemorrhage were 4.5 times more affected by asphyxia as compared to neonates who delivered from mothers who had no antepartum hemorrhage(AOR = 4.5, 95% CI, 2.3, 8.6). Furthermore, neonates from mothers with gestational diabetes mellitus were four times more likely to be asphyxiated than neonates from mothers without gestational diabetes mellitus (AOR = 4.2, 95% CI 1.9,9.2). Moreover, neonates of mothers who had a history of premature rupture of the membrane were 2.5 times more likely to be asphyxiated than those who had not encountered premature rupture (AOR = 2.5, 95% CI,1.33,4.7). Among the fetal factors, fetal distress during labor and meconium-stained

**Table 2. Obstetrics history of mothers at WCSH, South central Ethiopia (n = 311).**

| Variable | Category | Frequency(n = 311) | Percentage (%) |
|---|---|---|---|
| Parity | Primipara | 115 | 37% |
| | Multipara | 196 | 63% |
| Gestational age | <37weeks | 43 | 13.8% |
| | >42weeks | 18 | 5.8% |
| | 37week-42week | 250 | 80.4% |
| ANC[a] follow up | Yes | 197 | 63.3% |
| | No | 114 | 36.7% |
| Anemia | Yes | 129 | 41.5% |
| | No | 182 | 58.5% |
| Preeclampsia/eclampsia | Yes | 86 | 27.7% |
| | No | 225 | 72.3% |
| Chronic hypertension | Yes | 21 | 6.8% |
| | No | 290 | 93.2% |
| APH[b] | Yes | 92 | 29.6% |
| | No | 219 | 70.4% |
| DM | Yes | 24 | 7.7% |
| | No | 287 | 92.3% |
| Gestational DM[c] | Yes | 61 | 19.6% |
| | No | 250 | 80.4% |
| Chronic illness | Yes | 247 | 79.4% |
| | No | 64 | 20.6% |
| Mode of delivery | SVD[d] | 222 | 71.4% |
| | C/S[e] | 72 | 23.2% |
| | Instrumental | 17 | 5.5% |
| Duration of labor | >12hr(prolong) | 95 | 30.5% |
| | <12hr | 216 | 69.5% |
| PROM[f] | Yes | 100 | 32.2% |
| | No | 211 | 67.8% |
| Obstructed labor | Yes | 115 | 37% |
| | No | 196 | 63% |

[a]Antenatal Care

[b]Antepartum Hemorrhage

[c]Diabetes mellitus

[d]Spontaneous Vaginal Delivery

[e]Cesearian Section

[f]Prematue Rupture of Membrane.

amniotic fluid were significantly associated with asphyxia (AOR = 3, 95%CI, 1.3.7.0) and (AOR = 7.7, 95%CI, 3.1, 19.3) respectively [**Table 4**].

## Discussion

The findings from this study revealed that the magnitude of perinatal asphyxia was 41.2%, which is quite higher compared to a study done in southern India (2.7%) [29], Sweden (5.4%) [30], Alberta Canada (2.28%) [31] and South-East Nigeria (12.8%) [32]. This discrepancy could be explained by the socioeconomic variation between the study area and the other countries. Better developed countries have the necessary infrastructure and skilled health care

**Table 3. Clinical characteristics of neonates at WCSH, South central Ethiopia (n = 311).**

| Characteristics | Category | Frequency(n = 311) | Percentage (%) |
|---|---|---|---|
| Presentation | Vertex | 251 | 80.7% |
|  | Non vertex | 60 | 19.3% |
| Place of delivery | Home | 10 | 3.2% |
|  | Health center | 157 | 50.5% |
|  | Private clinic | 3 | 1% |
|  | Hospital | 141 | 45.5% |
| Gender | Male | 192 | 61.7% |
|  | Female | 119 | 38.3% |
| Birth weight | <2.5kg | 59 | 19% |
|  | 2.5kg-4kg | 240 | 77.2% |
|  | >4kg | 12 | 3.9% |
| Intrauterine meconium release | Yes | 84 | 27% |
|  | No | 227 | 73% |
| Fetal distress | Yes | 119 | 38.3% |
|  | No | 192 | 61.7% |
| Perinatal asphyxia | Yes | 128 | 41.2% |
|  | No | 183 | 58.8% |

providers to significantly reduce the incidence of perinatal complications. Nonetheless, the finding is lower than a study done in Bangladesh (56.9%) [33] and Ghana 283(61.8%) [34]. The possible explanation for this variation stems from the latter two studies being conducted in more poorly resourced areas. Additionally, the higher magnitude in these studies may be due to the reported low number of healthcare professionals trained to conduct neonatal resuscitation. Comparably, this study's result was higher than other studies conducted elsewhere in Ethiopia [12, 23, 28, 35, 36]. The difference could be attributed to the relative urban nature of the cities, which in turn affects the quality of setup amongst the facilities, the living standards of resident mothers and accessibility of health institutions. These are critical indicators that more emphasis is needed in studying perinatal asphyxia.

The Ethiopian ministry of health adopted the WHO recommendation of at least four ANC follow-up visits throughout a pregnancy. Despite this, ANC is utilized well in urbanized cities like Addis Ababa and Dire Dawa while semi urban and rural areas are far behind [22]. This is attributed to maternal illiteracy and low socio-economic status in the latter as these factors have direct relations to the delay in health seeking behavior [25]. The non-adherence to follow up could be catastrophic as it could endanger the maternal and fetal outcomes. This study revealed that 36.7% of mothers of neonates with asphyxia were not strictly attending their antenatal follow-up, paving the way for the 'silent' progress of pregnancy-induced complications (PICs) and associated perinatal outcomes.

The unpredictable nature of PICs, poor obstetric care and low service utilizations in Ethiopia, largely give rise to the maternal and perinatal sequelae. Even though the country adopted the "three delays" model, a set of strategies targeting delays in the decision to seek care, delays in seeking care, and delays in receiving adequate health care, successful implementation yet needs significant work [25]. This study has identified major PICs including preeclampsia, antepartum hemorrhage, gestational diabetes mellitus, premature rupture of the membrane that continue to be significant predictors of perinatal asphyxia.

Preeclampsia and eclampsia are known predictors of maternal and perinatal morbidities and mortalities [14]. These disorders, categorized under Pregnancy Induced Hypertensive

**Table 4. Factors associated with perinatal asphyxia among neonates admitted at WCSH, South central Ethiopia (n = 311).**

| Variables | Asphyxiated | | COR 95%CI | AOR 95%CI | p-value |
|---|---|---|---|---|---|
| | No | Yes | | | |
| **Educational status** | | | | | |
| No formal education | 85 | 78 | 2.07(.98, 4.4) | 1.6(.5, 5.1) | 0.420 |
| Primary | 40 | 24 | 1.4(.6, 3.2) | 1.2(.4, 3.7) | 0.624 |
| Secondary | 31 | 14 | 1.0(.4, 2.6) | 1.2(.4, 3.5) | 0.607 |
| Higher education | 27 | 12 | 1 | 1 | |
| **Residence** | | | | | |
| Rural | 114 | 93 | 1.6(.985,2.63) | 1.0(0.4,2.3) | 0.911 |
| Urban | 69 | 35 | 1 | 1 | |
| **Preeclampsia/Eclampisa** | | | | | |
| Yes | **25** | **61** | **5.75(3.33, 9.93)** | **6.2(3.1,12.3)** | **0.000**[*] |
| No | 158 | 67 | 1 | 1 | |
| **Antepartum hemorrhage** | | | | | |
| Yes | **31** | **61** | **4.464 (2.656, 7.5)** | **4.5(2.3,8.6)** | **0.000**[*] |
| No | 152 | 67 | 1 | 1 | |
| **Gestational diabetes mellitus** | | | | | |
| Yes | **17** | **44** | **5.115(2.757,9.5)** | **4.2(1.9,9.2)** | **0.000**[*] |
| No | 166 | 84 | 1 | 1 | |
| **Mode of delivery** | | | | | |
| SVD[a] | 139 | 83 | 1 | 1 | |
| CS[b] | 39 | 33 | 1.417(.828,2.425) | 1.0(0.5,2) | 0.912 |
| Instrumental | 5 | 12 | 4.019(1.368,11.81) | 2.3(0.5,9.5) | 0.258 |
| **Duration of labor** | | | | | |
| >12hr | 49 | 46 | 1.534(.942,2.5) | 1.1(0.6,2.2) | 0.703 |
| <12hr | 134 | 82 | 1 | 1 | |
| **Presentation of the fetus** | | | | | |
| Vertex | 153 | 98 | 1 | 1 | |
| Non vertex | 30 | 30 | 1.561(.886,2.75) | 1.5(0.7,3.5) | 0.311 |
| **Premature rupture of membrane** | | | | | |
| Yes | **42** | **58** | **2.782(1.705,4.54)** | **2.5(1.33,4.7)** | **0.004**[*] |
| No | 141 | 70 | 1 | 1 | |
| **Obstructed labor** | | | | | |
| Yes | 52 | 63 | 2.442(1.522,3.92) | 1.9(1.0,3.8) | 0.053 |
| No | 131 | 65 | 1 | 1 | |
| **Fetal distress during labor** | | | | | |
| Yes | **59** | **60** | **1.854(1.164,2.95)** | **3.0(1.3, 7.0)** | **0.010**[*] |
| No | 124 | 68 | 1 | 1 | |
| **Intrauterine meconium release** | | | | | |
| Yes | **23** | **61** | **6.33(3.62,11)** | **7.7(3.1,19.3)** | **0.000**[*] |
| No | 160 | 67 | 1 | 1 | |
| **Birth weight** | | | | | |
| < 2.5kg | 40 | 19 | 1 | 1 | |
| 2.5kg-4kg | 137 | 103 | 2.105 (0.6, 7.39) | 2.67(0.6, 12.6) | 0.207 |
| >4kg | 6 | 6 | 1.33(.417, 4.24) | 2.904(0.7,12.2) | 0.146 |

[a]Spontaneous Vaginal Delivery

[b]Cesearian Section

[*]AOR = statistically significant at p<0.05.

Disorders (PIHD), are common concerns globally, even more in low-income countries like Ethiopia. Studies have shown that variations in the prevalence of PIHDs can be observed within a country. For instance, the SNNPR region where the study area is located is one of the poorly resourced areas of the country and has a higher percentage of PIHD than any other region in the country [14]. Despite the absence of novel approaches to exactly predict the occurrence of preeclampsia, complications like birth asphyxia could be lowered through frequent and consistent follow-up during pregnancy. This study reaffirmed that preeclampsia is significantly associated with perinatal asphyxia. Studies conducted in India [29], Bangkok Rajavithi hospital [37], Gusau Nigeria [38], and Tigray Ethiopia [23] are in line with this data.

Significant evidence reveals [15, 39, 40] that the occurrence of antepartum bleeding results in a decreased blood flow from the mother to the placenta with subsequent hypoxia to the fetus. Studies conducted in Ethiopia have also shown an increased incidence of perinatal asphyxia among mothers who encountered APH [15, 40]. This study also identified that antepartum hemorrhage was associated with perinatal asphyxia, consistent with the study conducted in Bangalore, Indonesia [41], and Accra Ghana [34].

Gestational diabetes mellitus was also a factor associated with perinatal asphyxia, and this result is in agreement with what was reported in Canada [31], Pakistan [13], and the study done in Sweden [30]. The possible explanations for these are the metabolic derangement responsible for the inadequate production of surfactants. In Ethiopia, health care providers screen and initiate counseling for lifestyle modification during the early stages of the diagnosis, followed by a prescribed insulin regimen. Nevertheless, effective implementation of these strategies once again is hindered [42, 43]. The current study found out that premature rupture of the membrane was significantly associated with perinatal asphyxia which is consistent with other findings from Pakistan [30, 44], Ghana [34], and Tanzania [45]. The similarity between the studies may be explained by the scientific evidence that links PROM to the incidence of oligohydramnios, resulting in possible infection and umbilical cord compression [46].

The present study identified fetal distress and meconium-stained amniotic fluid as significant factors impacting the occurrence of perinatal asphyxia. Comparable results were found from other parts of the world; Malawi [47], Thailand [37], Pakistan [44], and India [48]. Studies conducted in central, south, and northern parts of Ethiopia are also in line [24, 36, 49]. The rationale for the poor perinatal outcome in Ethiopia likely stems from sub-standard partograph utilization, poor intrapartum care, and ineffective neonatal resuscitation practices [50–54].

## Limitations

Even though the study generated pertinent data that can be used as a baseline, the results can only describe the context in WCSH. Furthermore, the study also did not include other lower health care facilities where deliveries were conducted. The hospital where the study was conducted is an institution to which pregnant women with complications are referred. This may have inflated the magnitude of the problem. This study utilized only the APGAR score to diagnose perinatal asphyxia. Sarnat scores and additional investigations, such as the arterial blood gas analysis for pH, were not considered. The nature of cross-sectional study design also has an impact in defining cause and effect relationships.

## Conclusion

The study revealed that a substantial percentage of neonates experienced perinatal asphyxia. Major PICs such as preeclampsia, APH, and gestational DM, PROM, and fetal factors like fetal distress and intrauterine meconium release were significantly associated with the occurrence

of perinatal asphyxia. Focus on early identification and timely treatment of perinatal asphyxia in hospitals should, therefore, be given priority. Effective implementation of community engaging approaches such as mobilization and health information dissemination would improve the utilization of ANC services among mothers. Furthermore, revitalizing maternal waiting homes in the area may mitigate the delays in seeking care and associated complications. The Ministry of Health and the hospital administrators need to devise strategies that strengthen the health care system, meliorate the obstetric care setup and ensure the adherence of health care providers to intrapartum and immediate postpartum protocols such as partograph utilization and newborn resuscitation. Researchers should focus on conducting implementation research to assess the relative impact of strengthening and revitalizing best practices.

## Acknowledgments

We are indebted to Worabe comprehensive specialized hospital for providing us permission to conduct the study. We would like to thank Addis Ababa University, College of Health Sciences, for the offered technical support. Finally, the authors are also thankful to the supervisors and data collectors.

## Author Contributions

**Conceptualization:** Seifu Awgchew Mamo, Girum Sebsibie Teshome, Tewodros Tesfaye.

**Data curation:** Seifu Awgchew Mamo, Girum Sebsibie Teshome, Abel Tibebu Goshu.

**Formal analysis:** Seifu Awgchew Mamo, Girum Sebsibie Teshome, Tewodros Tesfaye, Abel Tibebu Goshu.

**Funding acquisition:** Seifu Awgchew Mamo, Tewodros Tesfaye.

**Investigation:** Abel Tibebu Goshu.

**Methodology:** Seifu Awgchew Mamo, Girum Sebsibie Teshome, Tewodros Tesfaye.

**Software:** Girum Sebsibie Teshome, Tewodros Tesfaye, Abel Tibebu Goshu.

**Validation:** Seifu Awgchew Mamo.

**Writing – original draft:** Seifu Awgchew Mamo, Girum Sebsibie Teshome, Tewodros Tesfaye, Abel Tibebu Goshu.

**Writing – review & editing:** Seifu Awgchew Mamo, Girum Sebsibie Teshome, Tewodros Tesfaye, Abel Tibebu Goshu.

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
