## [Decision Letter · Decision Letter 0]

24 Jun 2021

PONE-D-21-06937

Perinatal asphyxia and associated factors among neonates admitted to a specialized public hospital in South Central Ethiopia: A retrospective cross sectional study

PLOS ONE

Dear Dr. Goshu,

Thank you for submitting your manuscript to PLOS ONE. After careful consideration, we feel that it has merit but does not fully meet PLOS ONE’s publication criteria as it currently stands. Therefore, we invite you to submit a revised version of the manuscript that addresses the points raised during the review process.

I am including the thoughtful comments by 2 reviewers. To significantly improve the message of your paper a major revision will be needed.

We look forward to receiving your revised manuscript.

Kind regards,

Barbara Wilson Engelhardt, MD

Academic Editor

PLOS ONE

Journal Requirements:

2. Please ensure you have discussed any potential limitations of your study in the Discussion, including study design, sample size and/or potential confounders.

Reviewers' comments:

Reviewer's Responses to Questions

**Comments to the Author**

1. Is the manuscript technically sound, and do the data support the conclusions?

Reviewer #1: Partly

Reviewer #2: Yes

2. Has the statistical analysis been performed appropriately and rigorously? 

Reviewer #1: I Don't Know

Reviewer #2: Yes

3. Have the authors made all data underlying the findings in their manuscript fully available?

Reviewer #1: Yes

Reviewer #2: Yes

4. Is the manuscript presented in an intelligible fashion and written in standard English?

Reviewer #1: Yes

Reviewer #2: Yes

5. Review Comments to the Author

Reviewer #1: The article “Perinatal asphyxia and associated factors among neonates admitted to a specialized public hospital in South Central Ethiopia: A retrospective cross-sectional study” by Mamo et al identifies the associated factors seen with perinatal asphyxia at one hospital in a low- and middle-income country. This is important information to have as LMICs are now beginning to initiate quality improvement programs. I applaud them for their efforts. I do think the manuscript can be improved. If the issues I note below are addressed, it will be a much stronger piece and deserves publication. Once rewritten, it needs a careful review of English syntax and grammar. I have made some corrections below, but some areas will need significant rewrites.

Abstract

Concise. Well-written and appropriate.

Line 29 – Edit to “Perinatal asphyxia continues to be a significant clinical concern around the world as the consequences can be devastating.”

Include that the charts were reviewed from a 3 year period

Introduction

Line 55 delete the word “only”

Rework 55-60. Try this:

56-60 Perinatal asphyxia, also called birth asphyxia, is one of the leading causes of neonatal deaths in the world following severe infections and prematurity (1, 4). It results from the loss of the blood supply or impairment of gas exchange to or from the fetus before, during, or after the birth process (5). Perinatal asphyxia may lead to severe metabolic acidosis, hypercarbia, progressive hypoxemia, neonatal encephalopathy, and multi-system organ failure, and even result in death (6-10).

Line 70 delete the word “staggering”

Line 71 Is this the name of a program? If so, it should all be capitalized. National Strategy for Newborn and Child Survival

Rework 77-82. Try this:

77-82 In order to improve the quality of care that is delivered within a country, one must first understand the problem and the issues that healthcare providers encounter. Improving care of the maternal-infant dyad can lead to a significant decrease in the rates of perinatal asphyxia (find a reference for this). In an effort to improve care in Ethiopia, we sought to understand the burden of perinatal asphyxia and early assessment practices. To do this, we evaluated cases of perinatal asphyxia and identified associated factors among neonates admitted to the neonatal intensive care unit of Worabe Comprehensive Specialized Hospital in southern Ethiopia

Methods

This whole section is a little unclear. Were all records reviewed? This is what it sounds like from the first few lines. Or, were only 311 records reviewed based on some sampling procedure? This is what it sounds like as you read on.

The section: “Sample size determination and sampling procedure” needs to be rewritten. I need more information about what is being seen at Worabe in order to know if this even makes sense. How many births occurred there during the three-year period? How many are admitted to the NICU over the three-year period? How did you randomly select which charts to review? It is all very vague.

The biggest thing that is lacking is how was perinatal asphyxia defined? I am worried there was no uniform definition and a Sarnat 1 baby is being included in with Sarnat 2 and 3 babies. While technically all three are encephalopathies (stages of it), outcomes are very different and your information would be strengthened if we knew the definition and the range of severity in your definition.

Results

Line 167 “developed respiratory distress during labor.” This seems to be referencing the baby. How does a baby develop respiratory distress during labor? Looking at Table 3, I believe you mean the baby developed perinatal asphyxia during labor.

Line 168 – where does it say this in the table?

Line 172,173 – the table says it is 119 and 38.3%

Discussion

Line 225 – “the definition of birth asphyxia.” This is key. What is the definition for this study? I didn’t see it defined.

Overall, I think the discussion could use significant work rather than just stating that you found what others did. I would expect that to happen as these are all well-documented factors associated with perinatal asphyxia.

What I would really like to see done in this section is how could this information be applied in Ethiopia to improve care? How could you use this information to design an educational intervention and then measure the impact? Don’t just draw conclusions from the data, but let it lead you somewhere.

I also need you to think about what the limitations of your study might be. One I see are the potential problems with your data. Was it good? What about how the clinicians defined perinatal asphyxia? I am worried that your numbers are so high because of a variation in diagnosis amongst the providers.

Conclusion

Based on the above feedback, the conclusion would also need to be reworked.

Reviewer #2: Perinatal asphyxia and associated factors among neonates admitted to a specialized public hospital in South Central Ethiopia: A retrospective cross-sectional study.

Thank you for the opportunity to review this manuscript.

Below are comments/suggestions for the authors' consideration.

The article is well written, highlights an actual problem ( perinatal asphyxia ), and uncovers factors associated with perinatal asphyxia.

The abstract is concise and accurately summarizes the essential information of the paper.

The introduction is appropriate for the content of the article although it would be better if the authors include a paragraph to detail the

(i) background introduction information of Ethiopia relevant to this topic.

(ii) associated risk factors to perinatal asphyxia

The methodology can be further enhanced:

-- The study was conducted at Worabe Comprehensive Specialized Hospital (WCSH), south-central Ethiopia, perhaps the reader would appreciate it if the authors could detail the study setting.

-- The authors have mentioned the systematic random sampling method. (page 5, line 106), this should be elaborated further on the systematic random sampling method used in Data Collection Procedure. ( Page 5, line 115-119)

--Description of the checklist.

--May need to include the IRB ethical approval number in the text.

--Number of records that meet the inclusion criteria. & the number and reasons of records excluded in this study.

Data analysis procedures are sufficiently described. Results are organized in a way that is easy to understand. The statistics are reported appropriately. Corrections need for :

The total number of occupational statuses is more than 311. (Table 1)

Please add notes explaining any acronyms or abbreviations in the table. (Table 2 & Table 4)

Revise no illness as it is misleading ( Table 2)

The author has reported the frequency of mothers of asphyxiated neonates is 128 ( Table 1). This is incongruent with Table 3 Clinical characteristics of neonates with perinatal asphyxia, the researcher has reported 119 in the yes category. Please recheck.

In reporting of factor associated with perinatal asphyxia, it would be more meaningful to report

e.g ...neonates from mothers who had antepartum hemorrhage were 4.5 times more affected by asphyxia as compared to neonates who delivered from mothers who had no antepartum hemorrhage(AOR= 4.5, 95%CI, 2.3,8.6) (Page 10 . line 188-189)

Table 4: Please add in notes * p<0.05

The discussion and conclusion are well articulated. However, it was rather scanty. Suggest to elaborate further on “ This major difference could be attributed to the methodological approaches employed amongst the studies, the definition of birth asphyxia, and the management protocol of the hospitals for perinatal asphyxia. (Page 13, line 225-226) & as appropriate. And the study limitations are not discussed.

However, the references/ citations are appropriate.

Thank you

6. PLOS authors have the option to publish the peer review history of their article (what does this mean?). If published, this will include your full peer review and any attached files.

Reviewer #1: No

Reviewer #2: No

---

## [Author Response · Author response to Decision Letter 0]

31 Aug 2021

Responses for the Academic editor

Comment 1: Please ensure that your manuscript meets PLOS ONE's style requirements, including those for file naming.

Response 1: We respectfully accepted the comment. The submitted version is corrected according to the PLOS ONE's style requirements.

Comment 2: Please ensure you have discussed any potential limitations of your study in the Discussion, including study design, sample size and/or potential confounders.

Response 2: We respectfully accepted the comment. The revised version added the limitations of this study after the discussion section.

Comment 3: Please provide additional details regarding participant consent. In the ethics statement in the Methods and online submission information, please ensure that you have specified (1) whether consent was informed and (2) what type you obtained (for instance, written or verbal, and if verbal, how it was documented and witnessed). If your study included minors, state whether you obtained consent from parents or guardians. If the need for consent was waived by the ethics committee, please include this information.

Response 3: We respectfully accepted the comment. The anonymity of the study participants was kept confidential throughout the entire process. We have also added a statement pertaining to the comments under the section ‘Ethical considerations’.

Responses for Reviewer 1

First of all, we are very grateful for your encouraging words. We are humbled by the constructive feedback and the details you went through to review our work.

Responses for Reviewer 1 (Abstract)

Comment 1: Line 29 – Edit to “Perinatal asphyxia continues to be a significant clinical concern around the world as the consequences can be devastating.”

Response 1: We respectfully accepted the comment. It was edited on the revised document. 

Comment 2: Include that the charts were reviewed from a 3 year period.

Response 2: We respectfully accepted the comment. It was edited on the revised document. 

Responses for Reviewer 1 (Introduction)

Comment 3: Line 55 delete the word “only”

Response 3: We respectfully accepted the comment. It was deleted on the revised document.

Comment 4: Rework 55-60. Try this: 56-60 Perinatal asphyxia, also called birth asphyxia, is one of the leading causes of neonatal deaths in the world following severe infections and prematurity (1, 4). It results from the loss of the blood supply or impairment of gas exchange to or from the fetus before, during, or after the birth process (5). Perinatal asphyxia may lead to severe metabolic acidosis, hypercarbia, progressive hypoxemia, neonatal encephalopathy, and multi-system organ failure, and even result in death (6-10).

Response 4: We respectfully accepted the comment. It was edited as such on the revised document.

Comment 5: Line 70 delete the word “staggering”

Response 5: We respectfully accepted the comment. It was deleted from the revised document.

Comment 6: Line 71 Is this the name of a program? If so, it should all be capitalized. National Strategy for Newborn and Child Survival

Response 6: We respectfully accepted the comment. Yes, it is a name of a program and a correction was made.

 Comment 7: Rework 77-82. Try this: 77-82 In order to improve the quality of care that is delivered within a country, one must first understand the problem and the issues that healthcare providers encounter. Improving care of the maternal-infant dyad can lead to a significant decrease in the rates of perinatal asphyxia (find a reference for this). In an effort to improve care in Ethiopia, we sought to understand the burden of perinatal asphyxia and early assessment practices. To do this, we evaluated cases of perinatal asphyxia and identified associated factors among neonates admitted to the neonatal intensive care unit of Worabe Comprehensive Specialized Hospital in southern Ethiopia

Response 7: We respectfully accepted the comment. It was edited as such on the revised document.

Responses for Reviewer 1 on Methods

Comment 8: This whole section is a little unclear. Were all records reviewed? This is what it sounds like from the first few lines. Or, were only 311 records reviewed based on some sampling procedure? This is what it sounds like as you read on.

Response 8: We respectfully accepted the comment and question. The records that were reviewed were 311 out of the total neonatal records (3796) from the three years period. We used systematic random sampling technique to select the records. This was incorporated in the manuscript to make it clear.

Comment 9: The section: “Sample size determination and sampling procedure” needs to be rewritten. 

Response 9: We respectfully accepted the comment. The sub-section is rewritten with more details.

Comment 10: I need more information about what is being seen at Worabe in order to know if this even makes sense. How many births occurred there during the three-year period? How many are admitted to the NICU over the three-year period? How did you randomly select which charts to review? It is all very vague.

Response 10: We respectfully accepted the comment and questions. The total birth at Worabe hospital during the three years period was 10,800 and 3796 newborns were admitted to the NICU within this time frame. Systematic random sampling technique was used to select which charts to review. After the card records of the neonates were put in their order of admission, the kth interval was determined by dividing the total population size by the total sample size.

Comment 11: The biggest thing that is lacking is how was perinatal asphyxia defined? I am worried there was no uniform definition and a Sarnat 1 baby is being included in with Sarnat 2 and 3 babies. While technically all three are encephalopathies (stages of it), outcomes are very different and your information would be strengthened if we knew the definition and the range of severity in your definition.

Response 11: We respectfully accepted the comment and questions. In our study, perinatal asphyxia was defined as the inability of a newborn to initiate and sustain respiration, by scoring an APGAR score less than 7 persistently for more than 5 min after delivery.

Responses for Reviewer 1 on Results

Comment 12: Line 167 “developed respiratory distress during labor.” This seems to be referencing the baby. How does a baby develop respiratory distress during labor? Looking at Table 3, I believe you mean the baby developed perinatal asphyxia during labor.

Response 12: We respectfully accepted the comment and questions. It was an error during the preparation of the manuscript. The phrase on line 167 “developed respiratory distress during labor” was to mean “fetal distress”. What was presented in Table 3 also is corrected as “fetal distress” instead of perinatal asphyxia. 

Comment 13: Line 168 – where does it say this in the table?

Response 13: We respectfully accepted the comment. The percentage of neonates with and without perinatal asphyxia was incorporated into Table 3. The magnitude of perinatal asphyxia was indeed 41.2%. These errors are corrected on the revised manuscript.

Comment 14: Line 172,173 – the table says it is 119 and 38.3%

Response 14: We respectfully accepted the comment. It was an error and corrected as per the edition made to Table 3.

Responses for Reviewer 1 on Discussion

Comment 15: Line 225 – “the definition of birth asphyxia.” This is key. What is the definition for this study? I didn’t see it defined.

Response 15: We respectfully accepted the comment. The operational definitions for this study are added to the revised document.

Comment 16: Overall, I think the discussion could use significant work rather than just stating that you found what others did. I would expect that to happen as these are all well-documented factors associated with perinatal asphyxia.

Response 16: We respectfully accepted the comment. We added evidence-based supplements to the discussion section to modify it.

Comment 17: What I would really like to see done in this section is how could this information be applied in Ethiopia to improve care? How could you use this information to design an educational intervention and then measure the impact? Don’t just draw conclusions from the data, but let it lead you somewhere.

Response 17: We respectfully accepted the comment. The section was modified to address the concerns raised and it was contextualized based on the comments given.

Comment 18: I also need you to think about what the limitations of your study might be. One I see are the potential problems with your data. Was it good? What about how the clinicians defined perinatal asphyxia? I am worried that your numbers are so high because of a variation in diagnosis amongst the providers.

Response 18: We respectfully accepted the comment. The revised version incorporated the limitations of this study after the discussion section.

Responses for Reviewer 1 on Conclusion

Comment 19: Based on the above feedback, the conclusion would also need to be reworked.

Response 19: Thank you for the comment. The conclusion is rewritten in accordance with the discussion.

Thank you again!

Responses for Reviewer 2

First of all, we are very grateful for your encouraging words. We are humbled by the constructive feedback and the details you went through to review our work.

Responses for Reviewer 2 on Introduction

Comment 1: The introduction is appropriate for the content of the article although it would be better if the authors include a paragraph to detail the

(i) background introduction information of Ethiopia relevant to this topic.

(ii) associated risk factors to perinatal asphyxia

Response 1: We respectfully accepted the comment. These components are incorporated in the revised manuscript.

Responses for Reviewer 2 on Methods

Comment 2: The study was conducted at Worabe Comprehensive Specialized Hospital (WCSH), south-central Ethiopia, perhaps the reader would appreciate it if the authors could detail the study setting.

Response 2: We respectfully accepted the comment. We have included additional data pertaining to the study setting.

Comment 3: The authors have mentioned the systematic random sampling method. (page 5, line 106), this should be elaborated further on the systematic random sampling method used in Data Collection Procedure. ( Page 5, line 115-119)

Response 3: We respectfully accepted the comment. We have segregated the sub-section “Sample size determination and sampling procedure” into two. We explained the details on how we used the systematic random sampling under the “Sampling procedure” sub-section.

Comment 4: Description of the checklist.

Response 4: We respectfully accepted the comment. We have tried to describe the checklists used in this study under Data collection tools and procedures.

Comment 5: May need to include the IRB ethical approval number in the text.

Response 5: We respectfully accepted the comment. We have incorporated the ethics approval number granted by School of Nursing and Midwifery Research Ethical Committee into the revised document.

Comment 6: Number of records that meet the inclusion criteria. & the number and reasons of records excluded in this study.

Response 6: We respectfully accepted the comment. The calculated sample size was 311 and all randomly selected records fulfilled the inclusion criteria. We have not excluded a randomly selected record because we have robustly searched for other sources of data (to trace and find any missing information) including card room registration book, referral paper, ward admission and discharge registration book and documented nursing care plans.

Responses for Reviewer 2 on Results

Comment 7: The total number of occupational statuses is more than 311. (Table 1)

Response 7: We respectfully accepted the comment. A correction is made to the revised document.

Comment 8: Please add notes explaining any acronyms or abbreviations in the table. (Table 2 & Table 4)

Response 8: We respectfully accepted the comment. Notes explaining the acronyms or abbreviations are put under Tables 2 and 4. 

Comment 9: Revise no illness as it is misleading (Table 2) 

Response 9: We respectfully accepted the comment. It is corrected as chronic illness on the revised manuscript.

Comment 10: The author has reported the frequency of mothers of asphyxiated neonates is 128 ( Table 1). This is incongruent with Table 3 Clinical characteristics of neonates with perinatal asphyxia, the researcher has reported 119 in the yes category. Please recheck.

Response 10: We respectfully accepted the comment. The percentage of neonates with and without perinatal asphyxia was incorporated into Table 3. The magnitude of perinatal asphyxia was indeed 128 (41.2%). These errors are corrected in the revised manuscript.

Comment 10: In reporting of factor associated with perinatal asphyxia, it would be more meaningful to report e.g ...neonates from mothers who had antepartum hemorrhage were 4.5 times more affected by asphyxia as compared to neonates who delivered from mothers who had no antepartum hemorrhage(AOR= 4.5, 95%CI, 2.3,8.6) (Page 10 . line 188-189)

Response 10: We respectfully accepted the comment. It is corrected on the revised manuscript.

Comment 11: Table 4: Please add in notes * p<0.05

Response 11: We respectfully accepted the comment. The value of significance is added to the revised manuscript.

Comment 12: The discussion and conclusion are well articulated. However, it was rather scanty.

Response 12: We respectfully accepted the comment. Based on the suggestion, more content is added to enrich the discussion.

Comment 13: Suggest to elaborate further on “This major difference could be attributed to the methodological approaches employed amongst the studies, the definition of birth asphyxia, and the management protocol of the hospitals for perinatal asphyxia (Page 13, line 225-226) & as appropriate.

Response 13: We respectfully accepted the comment. These points are plausibly elaborated in the revised document.

Comment 14: And the study limitations are not discussed. However, the references/ citations are appropriate.

Response 14: We respectfully accepted the comment. The revised version incorporated the limitations of this study after the discussion section.

Thank you again!

---

## [Decision Letter · Decision Letter 1]

15 Oct 2021

PONE-D-21-06937R1Perinatal asphyxia and associated factors among neonates admitted to a specialized public hospital in South Central Ethiopia: A retrospective cross sectional studyPLOS ONE

Dear Dr. Goshu,

Thank you for submitting your manuscript to PLOS ONE. After careful consideration, we feel that it has merit but does not fully meet PLOS ONE’s publication criteria as it currently stands. Therefore, we invite you to submit a revised version of the manuscript that addresses the points raised during the review process.Thanks for resubmittion of an improved version of your paper.

I am including reviewers comments plus some of my own.Please submit your revised manuscript by January1 of 2022. If you will need more time than this to complete your revisions, please reply to this message or contact the journal office at plosone@plos.org. Please include the following items when submitting your revised manuscript:A rebuttal letter that responds to each point raised by the academic editor and reviewer(s). You should upload this letter as a separate file labeled 'Response to Reviewers'.A marked-up copy of your manuscript that highlights changes made to the original version. You should upload this as a separate file labeled 'Revised Manuscript with Track Changes'.An unmarked version of your revised paper without tracked changes. You should upload this as a separate file labeled 'Manuscript'.

We look forward to receiving your revised manuscript.

Kind regards,

Barbara Wilson Engelhardt, MD

Academic Editor

PLOS ONE

Additional Editor Comments (if provided):

Dear Dr. Goshu,

Thanks for resubmittion of an improved version of your paper.

I am including reviewers comments plus some of my own:

1. For the reader an explanation of the 3 delays model would be helpful.

2. Please give more detailed information regarding possible different infections of the fetus and the neonate.

3. What umbilical cord compromises have occured in this patient sample?

4. What percent of reduction of neonatal mortality do you aim for, what is feasible?

5. Please explain the capabilities of the different nurseries, especially the NICU - levels of care 1-4.

6. Most importantly: Please review your paper, most of all the discussion, for unnecessary redundancies, length of text.

Sincerely

Barbara Engelhardt

Reviewers' comments:

Reviewer's Responses to Questions

**Comments to the Author**

1. If the authors have adequately addressed your comments raised in a previous round of review and you feel that this manuscript is now acceptable for publication, you may indicate that here to bypass the “Comments to the Author” section, enter your conflict of interest statement in the “Confidential to Editor” section, and submit your "Accept" recommendation.

Reviewer #1: All comments have been addressed

Reviewer #2: All comments have been addressed

2. Is the manuscript technically sound, and do the data support the conclusions?

Reviewer #1: Yes

Reviewer #2: Yes

3. Has the statistical analysis been performed appropriately and rigorously? 

Reviewer #1: Yes

Reviewer #2: Yes

4. Have the authors made all data underlying the findings in their manuscript fully available?

Reviewer #1: Yes

Reviewer #2: (No Response)

5. Is the manuscript presented in an intelligible fashion and written in standard English?

Reviewer #1: Yes

Reviewer #2: Yes

6. Review Comments to the Author

Reviewer #1: Much improved manuscript.

Very minor grammar to address.

Line 26 - delete "in the world" as this is repetitive.

Line 334 - change to read "Utilization of the APGAR alone to diagnoses perinatal asphyxia may not..."

Reviewer #2: The paper has been significantly improved after revising.

It covers an important topic and is well written. In Abstract (page 2 lines 29-30), the authors stated that “ ...studies on the prevalence of asphyxia in these countries are limited.” The above statement is incorrect. This is not new information. Many other similar studies have been conducted in Ethiopia and published. It is unclear how this study specifically contributes to the literature. Suggest double-checking the similarity report too.

• https://journals.plos.org/plosone/article?id=10.1371/journal.pone.0255488 Prevalence and risk factors associated with birth asphyxia among neonates delivered in Ethiopia: A systematic review and meta-analysis

• doi: 10.1016/j.heliyon.2020.e03793 Prevalence of perinatal asphyxia in East and Central Africa: systematic review and meta-analysis

• DOI: 10.1155/2020/4367248 Prevalence and Associated Factors of Perinatal Asphyxia in Neonates Admitted to Ayder Comprehensive Specialized Hospital, Northern Ethiopia: A Cross-Sectional Study

• doi: 10.1371/journal.pone.0226891 Birth asphyxia and its associated factors among newborns in public hospital, northeast Amhara, Ethiopia

• https://bmcpediatr.biomedcentral.com/articles/10.1186/s12887-021-02598-z Neonatal mortality among neonates admitted to NICU of Hiwot Fana specialized university hospital, eastern Ethiopia, 2020: a cross-sectional study design

• DOI: 10.1155/2018/5351010 Prevalence and Associated Factors of Perinatal Asphyxia among Neonates in General Hospitals of Tigray, Ethiopia, 2018

• doi: 10.2147/PHMT.S196265 Prevalence and associated factors of perinatal asphyxia among newborns in Dilla University referral hospital, Southern Ethiopia– 2017

Suggest consolidating the Introduction. The operational definition should be elaborated further in this section (not Methodology).

Methodology-A citation is needed for the study area and period (Page 4, line 97)

Please revise the inclusion criteria as it is similar to the source population. (page 5, line 105 & line 110). Please take note that “2016 to December 31, 2019” appeared three times in Methods.

Please remove this statement “The overall prevalence of perinatal asphyxia among neonates who were delivered at WCHS was found to be 128(41.2%) from Clinical Characteristics” (page11, lines 204-205). You have reported it under the magnitude of perinatal asphyxia (Page 12, lines 228-229).

The Discussion has improved. Suggest to re-write Limitation because it does not appear to be sound. The conclusion can be further enhanced by adding recommendations for future research.

Thank you

7. PLOS authors have the option to publish the peer review history of their article (what does this mean?). If published, this will include your full peer review and any attached files.

Reviewer #1: No

Reviewer #2: No

---

## [Author Response · Author response to Decision Letter 1]

5 Nov 2021

Responses for the Academic editor

Comment 1: For the reader an explanation of the 3 delays model would be helpful.

Response 1: We respectfully accepted the comment. It was briefly explained in the Discussion.

Comment 2: Please give more detailed information regarding possible different infections of the fetus and the neonate.

Response 2: We respectfully accepted the comment. We have noted that the statement put in the Discussion was ambigious. However the statemet was put as a rationale for the similarity between ours and other studies. We also have tried to correct it as such on the revised document.

Comment 3: What umbilical cord compromises have occured in this patient sample?

Response 3: We thank you for the question. We have noted that the statement put in the Discussion was ambigious. However the statemet was placed as a rationale for the similarity between ours and other studies. We also have tried to correct it as such on the revised document.

Comment 4: What percent of reduction of neonatal mortality do you aim for, what is feasible?

Response 4: We thank you for the question. In alignment with the sustainable development goals, we aim for a 35-40% reduction in the magnitude of perinatal asphyxia at WCHS by the end of the decade.

Comment 5: Please explain the capabilities of the different nurseries, especially the NICU - levels of care 1-4.

Response 5: We respectfully accepted the comment. It was briefly discussed in the Methods section, ‘Study area’ sub-section of the revised document.

Comment 6: Most importantly: Please review your paper, most of all the discussion, for unnecessary redundancies, length of text.

Response 6: We respectfully accepted the comment. We have tried to omit redundancies throughout the discussion section.

Thank you again!

Responses for Reviewer 1

Comment 1: Much improved manuscript.

Response 1: We are thankful once again for taking your time to refine our work.

Comment 2: Line 26 - delete "in the world" as this is repetitive.

Response 2: We respectfully accepted the comment. We have omitted the phrase.

Comment 3: Line 334 - change to read "Utilization of the APGAR alone to diagnoses perinatal asphyxia may not..."

Response 3: We respectfully accepted the comment. We have corrected it as per your recommendation. 

Thank you again!

Responses for Reviewer 2

Comment 1: The paper has been significantly improved after revising. It covers an important topic and is well written. 

Response 1: We thank you for the encouraging words and the time you have spent in refining the document.

Comment 2: In Abstract (page 2 lines 29-30), the authors stated that “ ...studies on the prevalence of asphyxia in these countries are limited.” The above statement is incorrect. This is not new information. Many other similar studies have been conducted in Ethiopia and published. It is unclear how this study specifically contributes to the literature. Suggest double-checking the similarity report too.

Response 2: We respectfully accepted the comment. As per your recommendation, a correction was made to the specific statement that indicated there is limited information on the topic. We also believe the study’s findings will add to the body of knowledge interms of illuminating the problem in the study area. Furthermore, the findings of this study will direct responsible federal and local stakeholders to design effective implementations to reduce the magnitude of perinatal asphyxia.

Comment 3: Suggest consolidating the Introduction. The operational definition should be elaborated further in this section (not Methodology).

Response 3: We respectfully accepted the comment. We added a paragraph on the diagnostic definitions of perinatal asphyxia in the Introduction. Further elaboration was added in the methods section, as there was a comment by other reviewer.

Comment 4: Methodology-A citation is needed for the study area and period (Page 4, line 97)

Response 4: We respectfully accepted the comment. A citation is added to the description of the study area.

Comment 5: Please revise the inclusion criteria as it is similar to the source population. (page 5, line 105 & line 110). Please take note that “2016 to December 31, 2019” appeared three times in Methods.

Response 5: We respectfully accepted the comment. We have tried to reduce the redundancies by compiling only the necessary information under the sub section ‘Study population’.

Comment 6: Please remove this statement “The overall prevalence of perinatal asphyxia among neonates who were delivered at WCHS was found to be 128(41.2%) from Clinical Characteristics” (page11, lines 204-205). You have reported it under the magnitude of perinatal asphyxia (Page 12, lines 228-229).

Response 6: We respectfully accepted the comment. As per your recommendation, the statement is removed in the revised editon. 

Comment 7: The Discussion has improved. Suggest to re-write Limitation because it does not appear to be sound.

Response 7: We respectfully accepted the comment. We have tried to modify the Limitation in the revised edition.

Comment 8: The conclusion can be further enhanced by adding recommendations for future research.

Response 8: We respectfully accepted the comment. We added recommendations for future researchers in the revised document. 

Thank you again!

---

## [Editor Report · Decision Letter 2]

31 Dec 2021

Perinatal asphyxia and associated factors among neonates admitted to a specialized public hospital in South Central Ethiopia: A retrospective cross-sectional study

PONE-D-21-06937R2

Dear Dr. Goshu,

We’re pleased to inform you that your manuscript has been judged scientifically suitable for publication and will be formally accepted for publication once it meets all outstanding technical requirements.

Thank you for sending the revised copy. Except for some minor changes/suggestions, which I highlighted, the paper is now complete.

I have uploaded the last version of your paper with my suggestions for change, which should be very easily and quickly done.

Kind regards,

Barbara Wilson Engelhardt, MD

Academic Editor

PLOS ONE

Additional Editor Comments (optional):

See upload from 12/29

.

.

.

---

## [Editor Report · Acceptance letter]

5 Jan 2022

PONE-D-21-06937R2 

Perinatal asphyxia and associated factors among neonates admitted to a specialized public hospital in South Central Ethiopia: A retrospective cross-sectional study 

Dear Dr. Goshu:

I'm pleased to inform you that your manuscript has been deemed suitable for publication in PLOS ONE. Congratulations! Your manuscript is now with our production department. 

Kind regards, 

on behalf of

Dr. Barbara Wilson Engelhardt 

Academic Editor

PLOS ONE